# An Environmental and Nutritional Evaluation of School Food Menus in Bahia, Brazil That Contribute to Local Public Policy to Promote Sustainability

**DOI:** 10.3390/nu14071519

**Published:** 2022-04-06

**Authors:** Alana Kluczkovski, Camilla A. Menezes, Jacqueline Tereza da Silva, Leticia Bastos, Rebecca Lait, Joanne Cook, Bruno Cruz, Bruna Cerqueira, Renata M. R. S. Lago, Alexvon N. Gomes, Ana Marice T. Ladeia, Ximena Schmidt Rivera, Nelzair Vianna, Christian J. Reynolds, Ricardo R. Oliveira, Sarah L. Bridle

**Affiliations:** 1Department of Physics and Astronomy, School of Natural Science, University of Manchester, Manchester M13 9PL, UK; beckielait@googlemail.com (R.L.); sarah@sarahbridle.net (S.L.B.); 2Department of Biology, University of York, Wentworth Way, York YO10 5DD, UK; 3Postgraduate Program in Human Pathology, Postgraduate Program in Biotechnology in Health and Investigative Medicine, Gonçalo Moniz Institute, Oswaldo Cruz Foundation, Fiocruz, Salvador 40296-710, Brazil; nutcamilla@gmail.com (C.A.M.); letybastosm@gmail.com (L.B.); ricardo.riccio@fiocruz.br (R.R.O.); 4Global Academy of Agriculture and Food Security, The University of Edinburgh, Edinburgh EH8 9YL, UK; tsilva.jacqueline@gmail.com; 5Department of Environment and Geography, University of York, Wentworth Way, York YO10 5DD, UK; jo2cook@gmail.com; 6School of Nutrition, Noble University Center, Feira de Santana 44001-008, Brazil; bruno-cruzz@hotmail.com (B.C.); bruna.cerqueira.t@gmail.com (B.C.); 7Postgraduate Program in Medicine and Human Health, Bahiana School of Medicine and Public Health, Salvador 40290-000, Brazil; lagorenata1973@gmail.com (R.M.R.S.L.); alexvongomes@yahoo.com (A.N.G.); anamarice@bahiana.edu.br (A.M.T.L.); 8Equitable Development and Resilience Research Group, Department of Chemical Engineering, College of Engineering, Design and Physical Science, Brunel University London, London UB8 3PH, UK; ximena.schmidt@brunel.ac.uk; 9Molecular Epidemiology and Biostatistics Laboratory, Gonçalo Moniz Institute, Oswaldo Cruz Foundation, Fiocruz, Salvador 21040-900, Brazil; nelzair.vianna@fiocruz.br; 10Centre for Food Policy, City University of London, Northampton Square, London EC1V 0HB, UK; c.reynolds@sheffield.ac.uk; 11Department of Geography, University of Sheffield, Sheffield S10 2TN, UK

**Keywords:** climate change, GHGE, plant-based diets, public school meals, nutrition, diet quality, children health, sustainable school program, healthy citizen, food provision

## Abstract

Aimed at improving the quality of school meals, the Sustainable School Program (SSP) implemented low-carbon meals, twice a week, in 155 schools of 4 municipalities, reaching more than 32,000 students. This study evaluated the environmental impact and nutritional viability of this intervention for this population. The 15 most repeated meals from the conventional and sustainable menus were selected, and we considered the school age group and number of meals served per student/day. Nutritional information was calculated using validated food composition tables, nutritional adequacy was assessed using National School Feeding Program (PNAE) requirements, the level of processing was considered using NOVA classification, and greenhouse gas emissions (GHGE) were estimated using food life cycle assessment (LCA) validated data. We found both conventional and sustainable food menus are equivalent, in terms of nutrients, except for calcium, iron, and magnesium. Sustainable food menus were cholesterol-free. However, there was a reduction of up to 17% in GHGE, depending on the school age group analysed. Considering the greater energy efficiency and lower environmental impact of these food menus, the SSP, therefore, demonstrates that a substantial reduction in climate impact is feasible, successful, and can be an inspiration to other regions globally.

## 1. Introduction

More than a quarter of the global greenhouse gas emissions (GHGE) are emitted by the food system [1], while malnutrition is one of the main sources of mortality in the world [2,3]. The increasing threat of climate change will likely affect agriculture, hence endangering food security [4]. Furthermore, rising sea levels and frequent flooding will adversely impact communities living close to the coastal area [4]. Additionally, malnutrition, due to lack of access and availability of affordable and culturally relevant nutritious food, could lead to higher consumption of cheap, low-quality, processed food, with high contents of unhealthy ingredients, such as sugar, salt, and fat, putting pressure on the health system, with non-communicable diseases requiring expensive and regular health treatments [5].

Sustainable food production needs to promote agricultural practices, not only with lower GHGE, but also those supporting local biodiversity, resource efficiency, and the welfare of the population, in terms of quality, affordability, and accessibility to the product, which is essential for countries to meet the United Nations Sustainable Development Goals (SDGs) [4,6].

Whilst current efforts and school curriculums aim at promoting actions that reduce the anthropogenic impacts on climate change, for example, through reducing vehicle use and turning out lights, research investigating the impacts of animal agriculture on climate change have shown how efforts focused on reducing climate-impactful foods in diets are the key to decrease GHGE [7] and can drastically overpower other common strategies (such as turning off lights, etc.). For instance, a reduction of 46% (primary energy demand) and 60% (blue water footprint) on environmental impacts was found [8].

The Brazilian National School Feeding Program (PNAE) is Brazil’s longest-standing public policy, promoting food security by contributing to the bio-psychosocial development and educational achievement of students, as well as by meeting their nutritional needs while in the classroom and supporting the formation of healthy habits through food and nutrition education [9]. The program’s large coverage and innovative design act also to strengthen family farming, while promoting access to adequate and healthy diets in all public and community schools in the basic education system, from day care, kindergarten, elementary school, and high school to education for young adults [10]. The program was created in a time when hunger and undernutrition was a main problem. Therefore, its main aim was to offer around 15% of children’s nutritional needs. In 2018, PNAE benefited 40 million students in Brazilian public schools [11]. If we add up the students from private schools, this number will be much higher. Therefore, acting at schools has the power to reach a broad population, with the possibility to implement long-term behaviour change. However, with the nutritional transition and the current global syndemic of undernutrition, obesity, and climate change, the role of the program should be rediscussed as so to offer meals that will face multiple challenges.

The Sustainable School Program (SSP), or “Programa Escola Sustentável”, is an initiative of the State Public Ministry of Bahia, whose main objective is to monitor school meals by fostering actions consistent with the right to adequate and inclusive food, health prevention, quality education, and protection of the environment. In 2018, a pilot program was implemented, with the objective of improving the quality of school meals, through the redesign of conventional menus and progressive adoption of menus constituted, preferably with ingredients of vegetable origin that are produced by the local rural family entrepreneurs. In 2019, sustainable meals were implemented, twice a week, in 155 schools of four municipalities in the semi-arid region, with low human development index, impacting more than 32,000 students.

This study assesses the potential environmental benefits of transforming school food menus from conventional to sustainable, all whilst ensuring that nutritional requirements were met. With schools being an environment rich in learning and personal development, it is paramount to ensure that the lessons taught, both inside and outside of the classrooms, are useful and engaging. The climate crisis has highlighted the social and environmental responsibility of those delivering the information, to not only appropriately convey the severity of the climate crisis, but to share the ways in which each individual can positively affect their level of impact on the environment. Therefore, this study has developed educational materials to fulfil this social and environmental responsibility by enabling the nutritionists and students to calculate the GHGE of the school meals, thus being a tool to provide continuous improvements of current and future food menus, as well as resources for food climate change education.

## 2. Materials and Methods

This section presents a detailed description of how school food menus were sampled, as well as the nutrition and environmental impact calculated. Descriptions of the development of educational resources and ethical aspects is also found below.

### 2.1. School Food Menus

In 2019, nutritionists prescribed and implemented food menus for students, from nursery, pre-school, elementary, secondary, young adults, and adults, in 200 school days, in the municipal school system of the following four cities in the state of Bahia: Barrocas, Biritinga, Serrinha, and Teofilândia. Due to the differences between the number of meals offered per day for the different groups of students, items that make up the menus, and quantities of *per capita* consumption, it was decided to analyse the menus separated into two categories: Group 1—nursery and pre-school (0 to 5 years old); Group 2—elementary, secondary, young adult, and adult education (from 6 years old). Menus were defined based on a sample consisting of the 15 most repeated meals of Group 1 (whose students make three meals per day at school) and 10 from Group 2 (whose students make two meals per day at school), which were selected from each menu entitled conventional that contained foods of animal origin (implemented three times a week), as well as the one entitled sustainable, exclusively composed of plant-based foods (implemented twice a week). These 30 meals of Group 1 and 20 meals of Group 2 were gathered on four menus, defined as: Group 1 conventional menu; Group 1 sustainable menu; Group 2 conventional menu; Group 2 sustainable menu; this enables the comparison of nutritional content and environmental impact, as well as the statistical analysis. A summary of the menus is displayed in Table 1.

### 2.2. Nutritional Content

The evaluation of the meals’ nutritional composition was conducted using the online software Dietbox^®^, using the Brazilian Institute of Geography and Statistics food composition table, prepared by the Brazilian Institute of Geography and Statistics [12] and completed with support for nutritional decision [13], in the case of missing items. To assess nutritional adequacy, the requirements described in Annex III of Resolution CD/FNDE No. 26 [14]. The resolution specifies the minimum daily offer of total calories, carbohydrates, proteins, lipids, fibres, vitamins A and C, calcium, iron, magnesium, and zinc in school meals. It also determines maximum levels allowed for saturated fat, sugar, and sodium. Despite not being determined or suggested by the legislation, it was decided to include the amounts of cholesterol and vitamin B12 in this analysis, using the reference values recommended by the Institute of Medicine (US)—Standing Committee on the Scientific Evaluation of Dietary Reference Intakes, as well as its panel on folate, other B vitamins, and choline [15,16].

A qualitative evaluation of the school meals was conducted using the NOVA classification system, which classifies foods into four groups, based on their level of processing [17]. Group 1 is composed of unprocessed and minimally processed foods, such as fruits, vegetables, legumes, cereals, pulses, milk, eggs, and meat. In this group, foods are consumed raw or require simple processing, such as cooking and pasteurization. Group 2 includes processed culinary ingredients, such as oils, butter, lard, sugar, and salt. In Group 3, they are processed foods, such as salted meat, fruit jellies, fish, grains, and canned vegetables. They are usually produced by adding Group 2 substances to Group 1 foods and using preservation methods, such as canning and bottling. Group 4 comprises of ultra-processed foods, which are formulations of ingredients, mostly for exclusive industrial use, produced through a series of industrial processes that include the fractionation of whole foods into isolated substances and frequent use of additives, to increase or reduce the content of a certain nutrients and improve the sensory characteristics of the final product. This group includes soft drinks, cookies, ice cream, margarine, sliced bread, etc.

### 2.3. Environmental Impact

From the analysis of school food menus, the amount of emissions from the conventional and sustainable menus were calculated on a school calendar year. To accurately calculate the GHGE values of the menus, it was essential to build a strong foundation by extracting the GHGE values of the individual ingredients from reliable sources. The databases used were [18,19]. These values were used to generate a table to display the relevant information for each food. Since the data provided in the scientific papers and databases was calculated for a generalised portion size, each value was adjusted to account for the specific size of each meal. GHGE were calculated in kilograms of carbon dioxide equivalent per year (kg CO_2_e/year). The results of these calculations allowed the emission values for both the conventional and sustainable menus to be compared, such that further analysis could be conducted to calculate the potential benefits in emission reductions, as a result of choosing a sustainable menu over a conventional version. In addition, this approach allows for scaling up the results and identifies the potential benefits of increasing the number of sustainable days at school, according to the SSP design.

### 2.4. Public Engagement Component

Educational resources were developed to inform and engage the school community, as well as to raise awareness on the impact food has on the environment. A booklet was created, which presented information about food sustainability; the step-by-step methodology of the GHGE calculations, including a table with the food ingredients analysed in the school food menus, as well as the GHGE values associated with each food. To accompany the booklet, a GHGE menu calculator, containing the calculation method, was created using the web-based Google Docs editor and Google Sheets, which was made available to the participant schools of the SSP. A tutorial video was created to explain how to use the booklet, along with the calculator.

### 2.5. Statistical Analysis

The nutritional content and impacts to climate change (GHGE), regarding the conventional and sustainable food menus, were described as median and interquartile range, separated for Groups 1 (nursery and pre-school) and 2 (elementary, secondary, young adult, and adult education). Differences between conventional and sustainable menus in each group were tested using the Kruskall-Wallis test. The analysis was performed using the R software, version 4.1.0 [20].

### 2.6. Ethical Aspects

This study is part of the research entitled “Evaluation of an intervention project in school meals on the health of children and adolescents in the hinterland of Bahia” (CAAE: 91282318.3.0000.5544), approved by the Ethics Committee in Research with Human Beings of Bahiana School of Medicine and Public Health, on September 17/2018, under the opinion of number 2.962.623, as determined by resolution CNS 466/2012.

## 3. Results

### 3.1. School Meals

Table 1 shows the four food menus, consisting of the most frequently repeated meals for each group, which were created to allow the comparative analysis of the nutritional content and GHGE among the conventional and sustainable recipe options.

To preserve the nutritional content of the meal, the sustainable menus replaced animal-based products (meats, dairy products, and eggs) with legumes (soya and peanuts). In both food menus, the cereals and tubers group varied between preparations, such as bread and cake (wheat), couscous (corn), rice, cassava, and yams. Additionally, the fruits and vegetables group contained both fresh fruits, juices, and smoothies, as well as raw and cooked vegetables. The sustainable food menu included preparations that represent a reinterpretation of those presented in the conventional food menu, such as the replacement of cow’s milk with peanut milk in the same preparations (smoothie and porridge). It also contained typical preparations of the local food culture, some examples were “Black beans casserole”, “Rice and black-eyed beans”, and “Sweet corn coconut pudding”, which are commonly accepted by children and young adults; there were also adaptations of classical meals, such as “soya mince sandwich” (a version of the hot dog) and “Sweet rice pudding”, made free of animal-based ingredients.

### 3.2. Nutritional Content and Qualitative Evaluation

As seen in Table 2, among the food menus for Group 1, the sustainable menu contains more total calories, lower protein, and more fibre than the conventional one, which is a common characteristic of plant-based diets. However, it also presented less carbohydrates, more added sugar, and provided more total fat (but less saturated fat); additionally, it was naturally cholesterol-free. Regarding micronutrients, the sustainable food menu has more iron, magnesium, and vitamin A, as well as less sodium, calcium, zinc, and vitamins C and B12. Among the food menus for Group 2, it was observed that the sustainable menu seemed to contain fewer total calories and protein (see Table 2). It also presented more carbohydrates and added sugar, as well as less total fat (but more saturated fat) than the conventional food menu, besides being naturally cholesterol-free and containing more fibre. Regarding micronutrients, the sustainable food menu presented less sodium and more magnesium, vitamin A, vitamin C, and iron; however, it also contained less calcium, zinc, and vitamin B12.

Table 2 presents the median, percentiles 25th and 75th, of the nutritional content of each food menu for different food groups. The only statistically significant differences (*p* value < 0.05) between the conventional and sustainable menus were found in the iron content for Group 1 and magnesium content for Group 2, with the sustainable food menu presenting higher levels (both *p* values = 0.047); in the case of calcium, the conventional food menu presented higher content in both groups (*p* value = 0.009). As expected, the other significant difference was observed in the cholesterol content (Group 1, *p* value = 0.005; Group 2, *p* value = 0.007), with higher levels on the conventional food menu because it is an animal origin fat and naturally free on plant-based menus. Data are shown in Table 2.

The sustainable food menu, planned for students of Group 1, presented a lower amount of calories from unprocessed foods and higher caloric share of processed and ultra-processed foods, when compared to the conventional one, as shown in Figure 1. On the other hand, in the food menus planned for Group 2, the situation is reversed; the sustainable had a higher caloric share of unprocessed foods, as well as a lower share of processed and ultra-processed foods. It is important to remark that, in all menus, for all groups, the unprocessed foods account for over 58% of the calories.

### 3.3. Nutritional Adequacy

According to the NSFP resolution, in force in 2019 [14], menus applicable to nursery and full-time pre-schools (Group 1 in this study), which include three meals a day, must offer at least 70% of the daily nutritional needs of students and, at most, 10% of total calories from saturated fat. For schoolchildren up to 3 years old, the addition of sugar to preparations is prohibited and, for those aged 4 years and over, the supply of added sugar is limited to 10% of total calories. Based on these criteria, it is observed in Figure 2 that both the conventional and sustainable menu for Group 1 met, and sometimes exceeded, the total calories needed of each student. In terms of carbohydrates, the sustainable menu met everyone’s demand, but the conventional was insufficient for those aged between 4 and 5 years old. Both menus reached the protein and total fat needed by all students; however, they exceeded the maximum acceptable levels of saturated fat. The same happened with the added sugar content. As it does not contain animal origin foods, the sustainable menu is naturally cholesterol-free; however, the conventional one was also adequate in this criterion for all age groups. For fibres, the conventional menu does not provide sufficient amounts for all age groups, and the sustainable one is below the needs for students between 4 and 5 years old.

Additionally, in accordance with the current legislation [14], menus applicable to elementary, secondary, youth, and adult education units (Group 2 in this study) that contain two meals a day must provide at least 30% of the daily nutritional needs of schoolchildren, with at most 10% of total calories from saturated fat and 10% from added sugar. Based on these criteria, Figure 2 demonstrates that, of the menus applied to Group 2, the conventional met the total calorie needs of almost all students, except those between 16 and 18 years old. The sustainable menu was only enough to meet the needs of students from 6 to 10 years old, falling for the other age groups. A similar situation occurred for proteins, with the conventional being sufficient to meet the demand of all students and sustainable only for those aged 6 to 10 years old. In terms of carbohydrates, both menus only met the needs of students aged 6 to 10 years, being insufficient for the other age groups. For total fat, both menus contemplated the needs of all age groups; however, they exceeded the maximum acceptable levels of saturated fat for all ages. The same happened with the added sugar content. The cholesterol content of the conventional menu was higher than acceptable for all age groups, while the sustainable menu was naturally free. For fibres, the conventional menu can be considered inadequate for all age groups, as well as the sustainable menu for students between 6 and 10 years old.

Figure 3 represents the adequacy of menus, in relation to the nutritional needs of students, in terms of micronutrients. For Group 1, i.e., students who eat three meals a day at school, the maximum sodium content allowed by current legislation for the menu is 1400 mg. It is possible to observe that both menus exceeded this limit for all age groups. The zinc content of the conventional menu met the needs of all students, while the sustainable one proved to be insufficient for those between 4 and 5 years old. For calcium, the conventional menu did not meet the needs of students aged 4 to 5 years, while the sustainable menu proved inadequate for those aged between 1 and 5 years. In terms of iron, the sustainable menu was sufficient to meet the needs of all students, while the conventional one was effective only for those aged between 1 and 3 years. Both menus met the needs of vitamins A, C, and B12, as well as magnesium, for all students, despite the absence of foods from the groups of meat, dairy, and eggs in the sustainable menu, which can be explained by the presence of fortified foods.

For Group 2, students who eat three meals a day at school, the maximum sodium content allowed by current legislation for the menu is 600 mg. Figure 3 shows that, equally, both menus exceeded the maximum sodium limits allowed for all age groups. The zinc content of the conventional menu met the needs of all students, while the sustainable one proved insufficient for all ages. For calcium, the conventional menu did not meet the needs of students aged 11 to 18 years old, while the sustainable one proved inadequate in all age groups. Both menus proved to be adequate, in terms of iron, vitamin A, and vitamin C in all age groups. Regarding the magnesium content, the sustainable menu was sufficient to meet the demands of students aged 6 to 15 years, while the conventional one proved to be even less efficient, serving only students aged 6 to 10 years. Both menus proved to be sufficient to meet the nutritional needs of vitamin B12 for the same reason previously described.

In general, it is possible to observe that both menus have limitations in meeting the recommendations for many nutrients, especially the minimum intake limits for protein, fibre, calcium, and zinc, as well as the maximum intakes for saturated fats, cholesterol, sugar, and sodium, being recommended adjustments to suit the target audience. A complete list of Nutritional targets for the menus according to age group is available on Appendix A.

### 3.4. Environmental Impact

The GHGE of the sustainable school food menus were lower, when comparing them with the conventional school food menus, for both age groups analysed in this study—Group 1: *p* = 0.047; Group 2: *p* = 0.036 (Figure 4). The percentage of sustainable days, where menus were implemented, increased progressively during the year 2019, as shown in Figure 5 (detailed data showing the total GHGE per year and the list of ingredients and *per capita* amount of menus are available on Appendix A). In the first semester, the sustainable food menu made up 20% of the total menus served to students in both groups. This amount increased to 40% in the second semester. Conventional food menus had a higher amount of GHGE, when compared to sustainable food menus, for both groups analysed (Figure 5). It is worth noting that, in 2018, the SSP only had planning actions for the implementation of school menus for the following year. Thus, in the first year of the program’s execution, the menus were still 100% conventional. A difference in absolute values was verified when analysing the emission values for Groups 1 and 2, with Group 1 having a higher value, due to a greater number of meals, compared to Group 2.

Following the original planning for progressive implementation of the SSP (which was not possible due to the COVID-19 pandemic), projections of the total emission values of the two types of food menus were calculated for both groups, as shown in Figure 5. The projections presented a decrease in the total amount of emissions, with the implementation of more sustainable days on the menus in both groups.

Figure 5 shows the contribution of sustainable food menus in the total GHGE per year, according to the implementation plan of the SSP, which progressively increases the percentage of sustainable food menus through the years. It started with 0% sustainable days to progressively reach 80%. It was observed that the total amount of GHGE decreased with the increase of sustainable food menus. As shown in Figure 5, adopting a sustainable menu four days a week reduces the GHGE from 400 kg CO_2_e/year to 240 kg CO_2_e/year for nursery and preschool menus, as well as from 242 kg CO_2_e/year to 132 kg CO_2_e/year for primary and secondary school menus.

The percentage decrease in the total GHGE produced per year, based on the implementation of more sustainable food menus, rather than conventional menus, is shown in Figure 6. By analysing these results, it was found that, the higher the use of the sustainable menus, the lower the greenhouse gas emission, for both school age groups. Moreover, starting from a 100% conventional menu in 2018 and moving to two days a week in 2019 reduced GHGE by 15% for school Group 1 and 17% for school Group 2. By increasing the ratio of the use of sustainable to conventional menus further, a reduction of 40% and 45% was to be expected in 2021, according to the school age group, due to the adoption of sustainable menus four days a week.

### 3.5. Public Engagement Component

As a part of disseminating information, and to positively engage with the school community, educational resources that focused on food sustainability and the calculation of GHGE were created. Datasets containing the environmental impacts of a diverse range of foods were carefully evaluated to extract their GHGE values and input them into an instructive booklet (Figure 7a).

Additionally, introducing the idea of global warming and effect on the climate to the reader, this booklet explained how food choices can have such an effect on this issue. An accompanying calculator (Figure 7b) and tutorial video were also created, in order to enable the public to independently calculate their own GHGE values for their meals. While the calculator (Figure 8a) allowed for the input of ingredients and their associated GHGE values, as found in the booklet, the tutorial video was designed to guide the public, step-by-step, on how to input this information into the calculator. Figure 8b shows the model spreadsheet, built in Portuguese, the language used in all educational materials produced in this study.

## 4. Discussion

School is considered a potential environment for health promotion, as well as for nutritional, environmental, and humanitarian education. School feeding, especially in Brazil, where it is supported by the largest and oldest public food and nutrition security policy in the country, is a strategic tool for promoting health, in addition to promoting environmental, economic, and social sustainability. According to the Food and Agriculture Organization (FAO), a sustainable diet should have a low environmental impact and contributing to high standards of food safety and health for future generations [21]. Of the nine planetary limits that need to be respected, in order to allow life on the planet, four of those that have already been exceeded are strongly related to food production and consumption. They are the biochemical flux of nitrogen and phosphorus, integrity of biodiversity, alteration of the earth system, and climate change. The ocean acidification limit is close to the uncertainty level [22]. Agricultural systems are responsible for 78% of the pollution of rivers and oceans, the use of freshwater in agriculture represents 70% of global water abstraction [23], and the food supply chain represents 26% of GHGE [19]. The reason for this impact seems to be the long food system, focused on the production of animals for consumption [24]. As vegetables represent a shorter production system, if coming from agroecological, local, and unprocessed production, plant-based diets can be considered more sustainable than diets containing animal products [25].

The concept of the global obesity, malnutrition, and climate change syndemic suggests that these three conditions may have the same origin, i.e., the unsustainability of food systems. While they contribute to the depreciation of natural resources, they also encourage unfair and unequal food distribution and provide the population with food of low nutritional quality, creating so-called “food deserts”, which contributes to food and nutrition insecurity globally. Data from the national Household Budget Survey reveal that Brazil follows the secular global trend of reduction in the prevalence of thinness and an increase in overweight and obesity in the school-age population [26]. More recently, the ERICA study [27] pointed to an even more alarming situation between Brazilian children and adolescents, i.e., the growing prevalence of comorbidities, such as metabolic syndrome (2.6%), hypercholesterolemia (20.1%), and systemic arterial hypertension (9.6%), even though specific nutritional deficiencies are still considered public health problems in this population.

According to the EAT–Lancet Commission, the path to feeding a future population of 10 billion people with a healthy diet, within the limits of the planet, requires that food production practices be revised, as well as that new food consumption patterns be encouraged. The suggestion is a reduction of at least 50% in the consumption of meat and sugar, as well as an increase of more than 100% in the consumption of vegetables by the year 2050 [2]. WHO, as part of its campaign to control childhood obesity, recommends, among other strategies, the participation of schools in promoting healthy eating habits [28]. The strategy promoted by the SSP, to optimize nutritional content and reduce GHGE, was to reduce the supply of meat, dairy products, and eggs, as well as to increase the supply of vegetables in school meals, implementing a menu entirely plant-based, once a week, during 2019, and predict a gradual enhancement of four times a week over a two-year period.

The results of the present study demonstrate that, despite this effort to promote acceptability and preserve nutritional viability, it was observed that the sustainable food menu, in general, has a lower frequency of fresh fruits and vegetables, as well as lesser diversity of foods and preparations. Additionally, a relevant aspect that hindered the comparative assessment between food menus, in terms of variety, is the difference in the format of meals, which did not follow a pattern (starter, main course, and dessert), and sometimes presented itself as a complete meal and sometimes as a snack, varying between one and three items, depending on the meal.

Regarding the nutritional aspects, sustainable menus seem to present more added sugar content than conventional ones. This could be partially explained because, in the sustainable menu, preparations such as yogurt (industrially sugared) or portions of fresh fruits (with natural sugar) were replaced by juice and smoothies, which mostly often require added sugar. A higher amount of total fat was also observed in the sustainable menu for Group 1. This fat profile could be due to the amount of peanut and soy recipes, which are rich in unsaturated fat. For Group 2, the sustainable menu presented more saturated fat, which may be explained by the use of recipes with coconut milk, such as sweet rice pudding and coconut cake, since coconut is rich in vegetable saturated fat. The lower content of vitamin B12 in the sustainable menus was expected, since this nutrient is found only in animal products and industrially fortified foods. Despite this, there was no statistically significant difference in the nutritional content between the preparations belonging to the conventional menu and those from the sustainable menu, except for the calcium, magnesium, iron, and cholesterol content.

Another important aspect is the presence of processed and ultra-processed foods in both menus, as the literature shows the risks of consumption to health [29,30]. Among animal products, processed meat has been identified as a major risk factor for the development of cancer in humans, so that limited evidence suggests that unprocessed red meat has a certain level of risk and is considered as a probable carcinogen in humans [31,32]. Although presenting processed and ultra-processed foods, the planned sustainable menus were free of animal origin foods from these groups. The conventional menu for Group 1 presented a higher frequency of foods from the group of meat, dairy, and eggs in unprocessed form than processed and ultra-processed. When replaced by foods such as soya and peanuts, which have a high protein concentration, with fewer calories than the meat group, it automatically reduces the percentage value of calories from unprocessed foods and increases the caloric participation of the other groups, unnecessarily representing a higher frequency of food groups. In Group 2 menus, the conventional meals presented processed and ultra-processed meats, in addition to unprocessed meats. When replaced by soya and peanuts, it automatically reduced the percentage value of calories from processed and ultra-processed foods and increased the caloric share of other groups.

Considering that one of the objectives of the SSP was the prevention of non-communicable chronic diseases, it is possible to suppose that the sustainable menu has a more interesting profile, since it is more effective in meeting fibre needs and respecting the maximum intake limits for saturated fat, cholesterol, and added sodium, when compared to the conventional menu. In addition to appearing more effective, from a nutritional aspect, the sustainable menu presented lower GHGE. This reduction highlights the impact changing foods has on climate change. This information provides an environmental evidence base to implement sustainable strategies and target where interventions have the most impact.

The menu designed to represent the conventional preparations most offered to students in Group 1 emitted an average of 1950 g CO_2_e/day, while the one, which represented the sustainable preparations, emitted 740 g CO_2_e/day. In the menus for Group 2, this reduction was from 940 to 420 g CO_2_e/day, similar to the results achieved by the model proposed for school feeding in Italy [33]. School menus from across the country were evaluated, and 194 preparations (70 starters, 83 main courses, 39 side dishes, 1 part fruit, and 1 part bread) were used to create sustainable menus, based on a mathematical model that associates nutritional adequacy, probable acceptability, and lower GHGE. The result was a 4-week menu, containing more vegetables, which were not only part of the side dishes, but of the starters and main courses, as well as less animal products, especially red meat. This menu would be able to meet the nutritional needs of the target audience (students from 6 to 11 years old) and has 525 g CO_2_e/day. Another mathematical model was devised to create a 4-week menu for Spanish elementary school, based on 2800 possible combinations of preparations, between 20 starters, 20 main courses, and 7 desserts, suggested by professional meal planners [34]. The result was a 20-day menu, with 15% lower cost and 24% less GHGE, when compared to those recommended by national guidelines. It is important to highlight that these studies created menus based on mathematical models, in order to serve as a basis for implementing intervention studies. They appear to be effective in reducing GHGE and costs, besides being nutritionally adequate, but intervention studies are needed to assess criteria such as acceptability and food waste.

In Stockholm, Sweden, an intervention in school feeding was carried out in three primary schools, with the implementation of a 4-week menu, containing two meals a day, totalizing 40 meals, in which six were plant-based. This menu emitted 497 g CO_2_e/day, while the previous one emitted 829 g CO_2_e/day (40% reduction). In addition, the optimized one was 14% lower cost, and there was no significant difference, regarding the acceptability and food waste criteria, between the pre- and post-intervention period [35]. A similar result was found in the experience that took place in the city of Barcelona, Spain. To promote healthier and more sustainable diets, low-carbon meals were implemented in all schools and public canteens in the city. The meals contained seasonal, organic, and locally produced foods, in addition to a reduction in animal origin food (especially red meat) and ultra-processed products. Compared to conventional menus, there was a 53% reduction in GHGE. The intervention proposed by the SSP was responsible for the reduction of up to 17% of GHGE in 2019 and could reach 45%, if implemented four times a week. The Spanish experience has also shown reductions of 60% in water footprint, 46% in primary energy demand, and 48% in land use, in addition to increasing the nutritional content of meals by up to 47% [8].

Plant-based diets, if well planned, are nutritionally appropriate for all stages of the life cycle, and can contribute to the prevention and treatment of chronic, non-communicable diseases [36]. A study that evaluated 20 systematic reviews and meta-analyses of observational and intervention studies demonstrated that plant-based diets are more related to positive health outcomes, such as better lipid profile and body mass index, as well as less associated with negative outcomes, such as ischemic diseases, diabetes, and cancer [37]. The Brazilian Ministry of Health, through the Dietary Guidelines for the Brazilian Population, states that encouraging the consumption of animal origin food can bring risks to human and planetary health, and suggests a reduction in consumption [38]. Although the results of this study demonstrate that, except for calcium, there is no difference in the nutritional content between menus containing foods of animal and plant-based origin, they also pointed out the inadequacies in meeting the nutritional needs of students. In general, both menus need adjustments in total energy, as well as the distribution of macro and micronutrients to be considered suitable for the target audience. The greater offer of other legumes (besides soy and peanuts), as well as fruits, vegetables, and seeds, could bring more diversity to the menus and variety among preparations. However, changes in menus need to be linked to nutritional education actions and acceptance evaluation, in order to avoid food waste, since the literature shows low acceptance among Brazilian teenagers [39,40,41,42].

In this study, it was possible to create public engagement tools to promote nutritional and environmental education. However, it was not possible to assess the impact of implementing these tools in the school community, for example, in reducing food waste, which suggests future studies. Another limitation of this study was the evaluation of conventional and sustainable preparations, through the creation of menus that brought together the most repeated meals throughout the 2019 school year. A future study could also evaluate the menus implemented throughout the whole school year, as well as the food baskets offered to families, due to the suspension of face-to-face activities, during the COVID-19 pandemic.

The new NSFP resolution, implemented in 2020, brings some changes in the specifications of the nutritional characteristics of school meals. Among them is the mandatory supply of heme iron at least four times a week [43]. As this nutrient is only found in animal origin foods, this means that school meals can be plant-based only once a week, and other days should include food from the meat, dairy products, and eggs groups. Even so, as demonstrated by the studies presented, it is possible to design school menus that meet nutritional recommendations and, at the same time, present lower GHGE.

It is essential that food guides include environmental aspects and advise society to adopt healthy practices that are within planetary limits, such as diets with a low carbon footprint, serving as a basis for promoting effective public policies to achieve the Sustainable Development Goals [44]. This issue has been discussed, since 2014, by the Dietary Guidelines for the Brazilian Population, which, despite including the group of meats, dairy products, and eggs, in the context of healthy eating, describes that the decrease in demand for animal origin foods significantly reduces GHGE, deforestation (resulting from the creation of new pasture areas), and intense use of water [38]. Planetary health has been considered a new discipline, and it should not be dissociated from human health by health professionals and public health policies. This is an imperative measure to fulfil the international agenda of 45% reduction in GHGE in 10 years, achieving the 17 Sustainable Development Goals by 2030 and carbon neutrality by 2050 [45].

The results of this work provided evidence to caterers and policymakers on how menus can be changed, in conjunction with effective co-developed education programmes, thus reducing the climate impact of food. This study is our first foray into calculating school menu climate impacts and, therefore, provides a pilot for our work in other settings. In addition, it could be usefully carried out in other regions of Brazil, as well as other countries.

## 5. Conclusions

This study has analysed the redesign of school food menus, from nutritional and environmental perspectives. The school food menus were similar. However, the conventional presented higher content of calcium and lower amounts of iron and magnesium. Another significant difference observed was the cholesterol content, with the sustainable menu being cholesterol-free. The environmental impact of the implemented sustainable food menus showed a reduction in GHGE. Projections for implementing more sustainable days showed a substantial reduction in GHGE. This highlights the importance of food choices, with regard to relieving humans’ impact on the climate crisis, as well as how we can effectively and appropriately deliver this information to children, in order to ensure the next generation of students are well-informed and knowledgeable about the importance of this topic. To empower and encourage students and nutritionists to calculate their own GHGE from food, educational materials (tutorial video and booklet) were produced. These resources ensure the dissemination of the program (and of this study), as well as the part that science plays in informing society of its findings. We have demonstrated that the sustainable diets programme has reduced total climate impacts by 15–17%. Due to the nutritional adequacy of the menus, they have considerable potential to reduce environmental impacts in Brazil and beyond.

## Figures and Tables

**Figure 1 nutrients-14-01519-f001:**
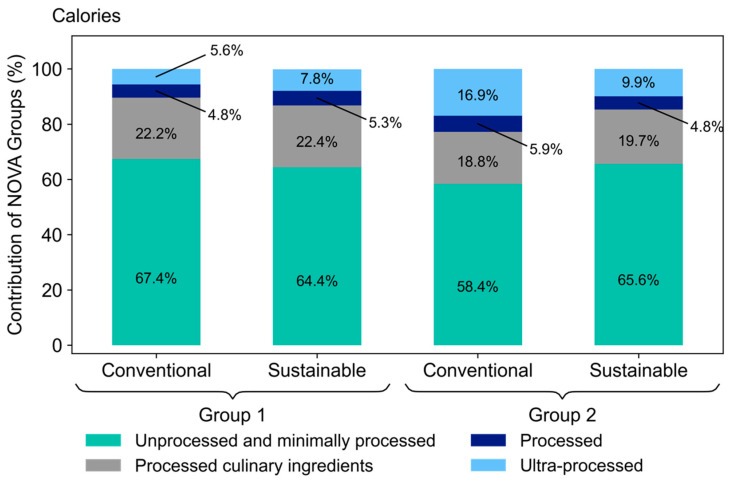
Share (%) of NOVA food groups to total calories in conventional and sustainable menus. Graph shows the contribution of NOVA food groups—unprocessed and minimally processed (teal), processed culinary ingredients (grey), processed (navy), and ultra-processed (light blue) food types; Group (1): nursery and pre-school; Group (2): primary, secondary, young adult, and adults.

**Figure 2 nutrients-14-01519-f002:**
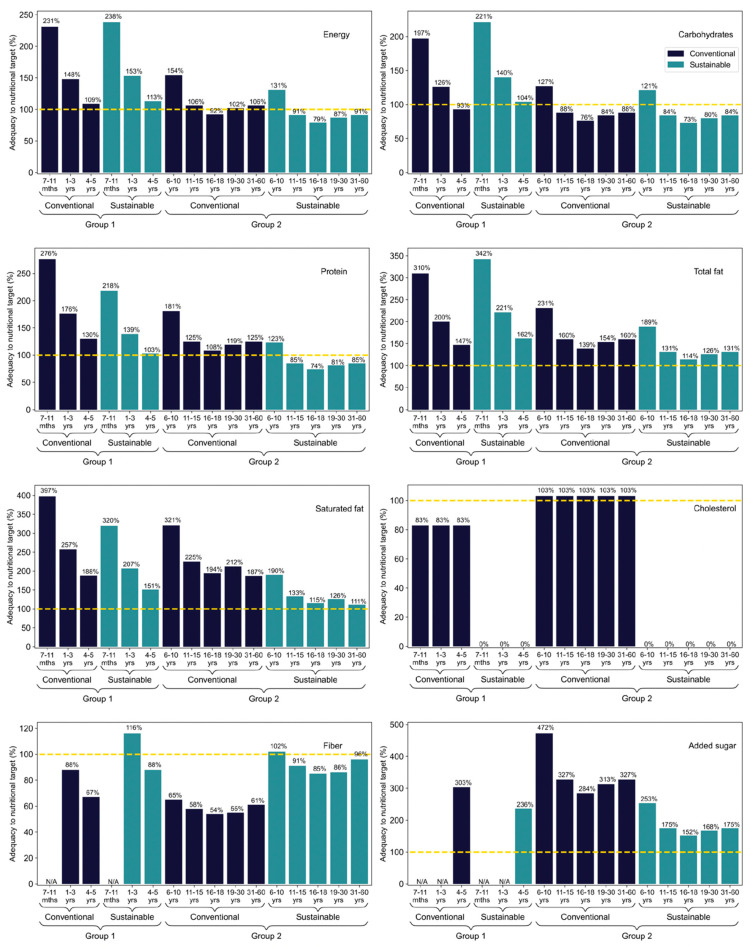
Macronutrient adequacy of conventional and sustainable menus to nutritional targets, according to the age group. Age groups considered were from 7 months to 60 years old, for both school food menus. Yellow dashed line represents the national daily nutritional recommendation for school meals (Group 1 is 70%, and Group 2 is 30% of the daily intake). For saturated fat, cholesterol, and added sugar, the daily nutritional recommendation refers to a maximum level. N/A means no official recommendations for this age group. Mths and yrs on the X axis mean months and years, respectively.

**Figure 3 nutrients-14-01519-f003:**
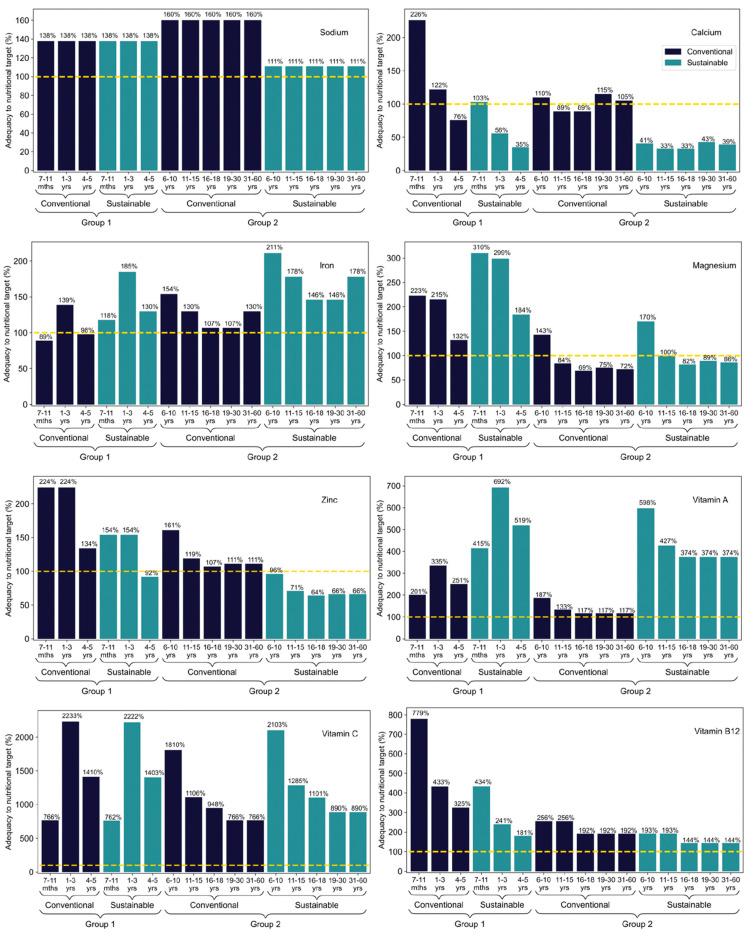
Micronutrients adequacy of conventional and sustainable menus to nutritional targets, according to the age group. Age groups considered were from 7 months to 60 years old, for both school food menus. Yellow dashed line represents the national daily nutritional recommendation for school meals (Group 1 is 70%, and Group 2 is 30% of the daily intake). For sodium, the daily nutritional recommendation refers to a maximum level. N/A means no official recommendations for this age group. Mths and yrs on the *X* axis mean months and years, respectively.

**Figure 4 nutrients-14-01519-f004:**
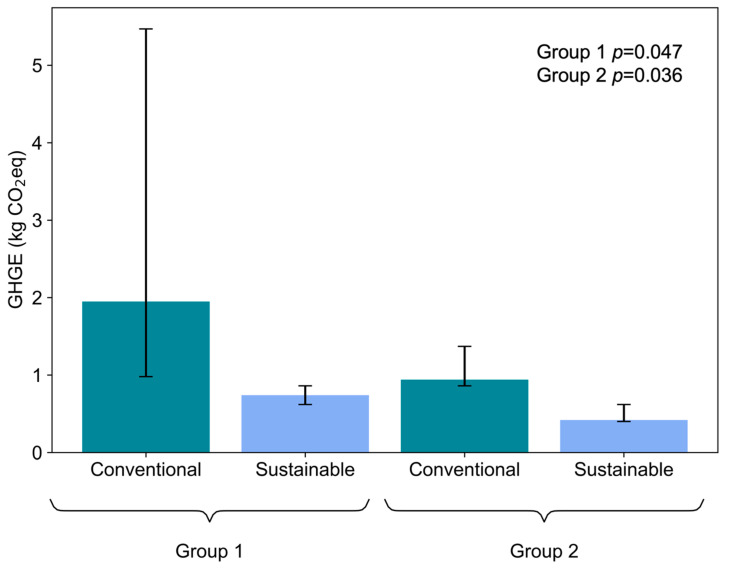
The GHGE value between sustainable and conventional school food menus. The GHGE expressed per year in kg CO_2_e for the two different food menu types targeted in this study. The graph shows the *p* value for both groups, and the error bars represent *p*25, *p*75 for each school food menu. See Section 2.1 for the classification of groups.

**Figure 5 nutrients-14-01519-f005:**
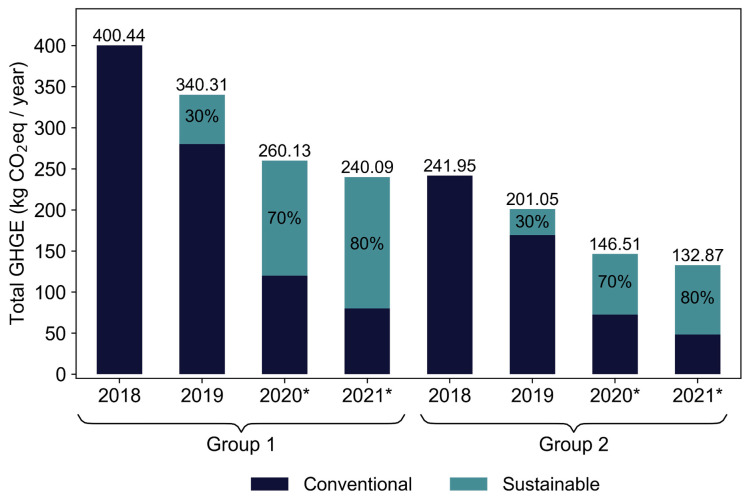
The total GHGE of school menus, including the contribution of sustainable and conventional food menus. The figure compares the estimated GHGE produced per year (in kg CO_2_e), between the two different food menu types targeted in this study. The percentages represent the amount of food menus being sustainable in each year, with 100% equivalent to 5 days a week. * The graph shows the projections for the years 2020 and 2021, following the implementation plan of the SSP; in 2018, there were no sustainable food menus.

**Figure 6 nutrients-14-01519-f006:**
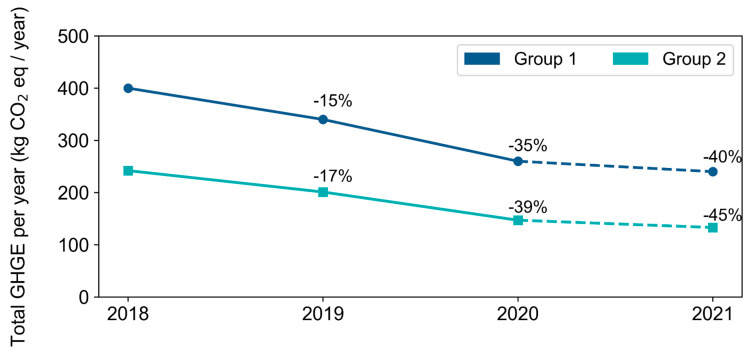
The percentage decrease in the total GHGE produced per year. The figure shows the total estimated GHGE produced per year in kg CO_2_e, based on the implementation of the specific ratios of conventional-to-sustainable food menus for both age groups. The graph shows projections for the years 2020 and 2021, following the implementation plan of the SSP.

**Figure 7 nutrients-14-01519-f007:**
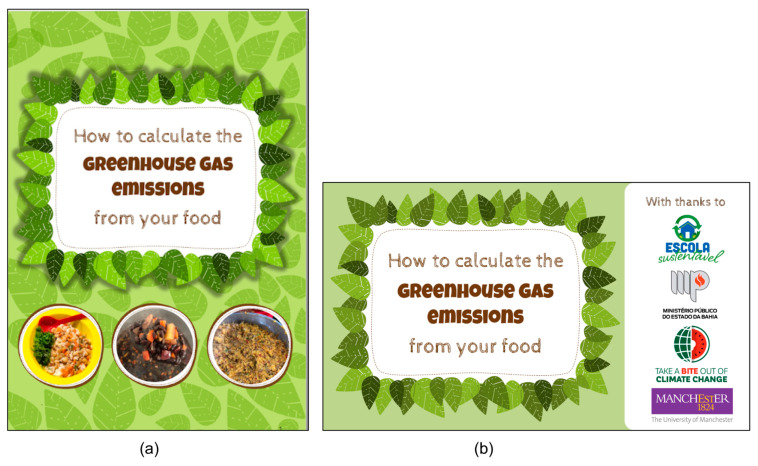
Educational resources developed in this study. Figure shows: (**a**) the booklet containing baseline information on GHGE, as well as tables of food ingredients and associated GHGE; (**b**) tutorial video explaining how to calculate GHGE from food.

**Figure 8 nutrients-14-01519-f008:**
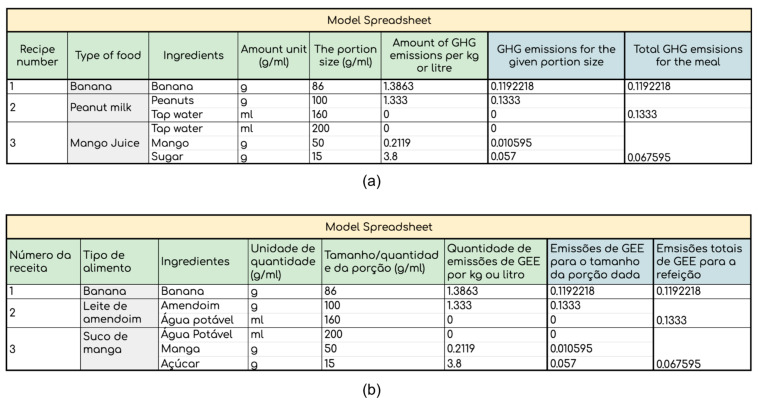
Educational calculator. Figure shows: (**a**) the calculator in Google spreadsheet in English, with the GHGE calculation example for banana, peanut milk, and creamy pudding; (**b**) the same spreadsheet in Portuguese.

**Table 1 nutrients-14-01519-t001:** Conventional and sustainable weekly menus, according to age group, selected for this analysis.

Day of the Week	Conventional	Sustainable
Group 1		
Monday	Meal 1: Latte + Sweet potato Meal 2: Lettuce and tomato salad + Pasta with sardine sauce + Carrot Meal 3: Mango juice + Carrot cake + Banana	Meal 1: Peanut milk + Cassava Meal 2: Beans + Pasta with soya mince + Potato with carrot + Guava juice Meal 3: Mango juice + Bread with peanut butter + Banana
Tuesday	Meal 1: Oatmal porridge + Papaya Meal 2: Rice + Beans + Chicken + Mashed potatoes + Guava Meal 3: Flavoured whole milk yogurt + Bread with margarine	Meal 1: Acerola juice + Sweet corn coconut pudding Meal 2: Lettuce salad + Rice + Beans + Okra and pumpkin soya chunks casserole Meal 3: Vegetable soup with soya mince
Wednesday	Meal 1: Acerola juice + Cassava with pulled beef Meal 2: Rice + Beans + Fish fillet with potato + Mango Meal 3: Latte + Sweet potato with egg + Papaya	Meal 1: Vegetable soup with soya mince + Finger rolls Meal 2: Rice + Couscous with vegetables and soya mince + Guava juice Meal 3: Papaya and apple smoothie + Cream cracker
Thursday	Meal 1: Latte + Couscous Meal 2: Rice + Beans + Minced beef with potato and carrot + Watermelon Meal 3: Guava juice + Coconut cookies	Meal 1: Banana and apple smoothie + Soya mince sandwich Meal 2: Rice + Black beans and vegetables casserole Meal 3: Plant-based Shepherd’s pie + Apple
Friday	Meal 1: Banana and apple smoothie Meal 2: Lettuce salad + Muleteer Beans with jerked beef + Braised kale Meal 3: Omelet + Apple	Meal 1: Banana and papaya smoothie + Coconut cake Meal 2: Lettuce and tomato salad + Rice and black-eyed beans with soya mince, carrot, pumpkin and kale Meal 3: Mango juice + Vegetable couscous
Group 2		
Monday	Meal 1: Vegetable and chicken soup + Banana Meal 2: Latte + Couscous with pepperoni	Meal 1: Beans and vegetable soup Meal 2: Vegetable couscous + Banana
Tuesday	Meal 1: Mashed potato + Beef bolognese pasta + Watermelon Meal 2: Hot chocolate + Cream cracker	Meal 1: Rice + Vegetables and soya chunks casserole Meal 2: Soya mince sandwich + Watermelon
Wednesday	Meal 1: Acerola juice + Cream cracker with guava jam + Papaya Meal 2: Latte + Sweet corn pudding	Meal 1: Acerola juice + Cream cracker with peanut butter Meal 2: Cornmeal porridge + Papaya
Thursday	Meal 1: Sweet rice pudding Meal 2: Pasta with tomato sauce + Diced chicken breast	Meal 1: Sweet rice pudding Meal 2: Mango juice + Soya mince + Bolognese pasta
Friday	Meal 1: Muleteer beans with jerked beef + Braised kale Meal 2: Latte + Bread with fried egg + Apple	Meal 1: Black beans and vegetables casserole + Cassava flour Meal 2: Guava juice + Coconut cake + Apple

Details of ingredients list and per capita consumption are in the Appendix A.

**Table 2 nutrients-14-01519-t002:** Nutritional content, as median (*p*25; *p*75), of the conventional and sustainable food menus, according to the age group.

Nutrient	Group 1			Group 2		
	Conventional	Sustainable	*p* Value	Conventional	Sustainable	*p* Value
**Energy, kcal**	1115.65 [897.49; 1171.88]	1146.15 [986.72; 1184.79]	0.754	713.47 [625.4; 748.58]	587.11 [524.64; 644.34]	0.117
**Carbohydrates, g/1000 kcal**	146.87 [123.85; 157.35]	144.94 [143.33; 161.72]	0.754	135.12 [115.16; 140.47]	151 [140.52; 175.65]	0.175
**Protein, g/1000 kcal**	32.55 [28.49; 42.14]	29.37 [26.16; 30.04]	0.175	37.02 [34.74; 43.11]	29.76 [29.06; 34.07]	0.117
**Total fat, g/1000 kcal**	29.46 [28.51; 39.72]	38.54 [32.17; 38.85]	0.465	37.58 [31.24; 45.63]	30.99 [30; 38.03]	0.602
**Saturated fat, g/1000 kcal**	12.6 [10.87; 16.44]	11.58 [8.87; 12.56]	0.347	11.74 [10.99; 15.46]	13.13 [8.38; 14.19]	0.602
**Cholesterol, mg/1000 kcal**	110.71 [102.64; 284.07]	0 [0; 0]	**0.005**	92.53 [76.17; 107.92]	0 [0; 0]	**0.007**
**Fiber, g/1000 kcal**	11.28 [8.99; 13.2]	15.26 [13.61; 16.28]	0.076	8.47 [3.69; 10.22]	10.81 [10.15; 21.67]	0.175
**Added sugar, g/1000 kcal**	25.6 [22.28; 40.34]	34.9 [29.54; 36.79]	0.754	23.98 [20.04; 24.92]	36.16 [0; 41.43]	0.916
**Sodium, mg/1000 kcal**	1908.22 [1612.69; 1912.33]	1830.43 [1723.08; 2087.48]	0.917	1930.95 [1764.69; 2037.05]	1069.16 [1065.83; 2002.03]	0.602
**Calcium, mg/1000 kcal**	342.92 [311.32; 478.6]	182.04 [180.87; 191.27]	**0.009**	461.94 [437.42; 533.8]	192.68 [171.79; 314.31]	**0.009**
**Iron, mg/1000 kcal**	6.24 [6.22; 6.85]	8.46 [7.74; 9.09]	**0.047**	7.31 [3.65; 8.31]	7.82 [7.37; 11.38]	0.295
**Magnesium, mg/1000 kcal**	111.32 [104.89; 126.39]	162.15 [152.4; 184.71]	0.076	112.48 [112.09; 126.22]	142.12 [130.45; 172.09]	**0.047**
**Zinc, mg/1000 kcal**	4.47 [4.2; 5.06]	2.71 [2.59; 3.16]	0.175	5.17 [3.18; 5.66]	2.04 [1.96; 2.27]	0.117
**Vitamin A, mcg/1000 kcal**	580.36 [560.69; 667.56]	1338.17 [1291.71; 1584.11]	0.076	214.14 [213.86; 272.96]	1841.13 [1188.3; 2167.75]	0.175
**Vitamin C, mg/1000 kcal**	118.61 [59.71; 180.11]	113.51 [95.84; 140.21]	0.754	45.91 [43.01; 74.5]	197.91 [84.23; 239.22]	0.347
**Vitamin B12, mcg/1000 kcal**	2.15 [1.58; 3.3]	1.15 [1.08; 1.53]	0.076	1.94 [1.6; 1.98]	1.63 [0.81; 2.98]	0.917

Group 1: Nursery and preschool; Group 2: Elementary, secondary and youth, and adult education. Numbers in bold show statistically significant differences (*p* value < 0.05).

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
