# Peer review of "An Environmental and Nutritional Evaluation of School Food Menus in Bahia, Brazil That Contribute to Local Public Policy to Promote Sustainability"

_nutrients, 2022, doi:10.3390/nu14071519_

Round 1

Reviewer 1 Report

It is suggested to modify figure 1 to simplify its interpretation: divide into Group 1 and Group 2 and inside indicate the Conventional and Sustainable histograms (such as figures 4 and 5).
Figures 2 and 3 are also complicated to interpret; groups 1 and 2 are not well identified.

Author Response

We are grateful for the reviewers’ comments and have been through them carefully. We believe the manuscript is significantly improved thanks to recommending their changes. We list below the ways we have modified the manuscript/figures in many places, and we explain for a couple of items why we feel they are beyond the scope of our paper to address fully at this time. We refer to the two reviewers as R1 and R2 in the below respectively.

R1: It is suggested to modify figure 1 to simplify its interpretation: divide into Group 1 and Group 2 and inside indicate the Conventional and Sustainable histograms (such as figures 4 and 5).

We thank the reviewer for the suggestion. We have improved Figure 1 to simplify the interpretation (indicate inside the conventional and sustainable menus), and grouped the items as suggested above. See change in page 08.

R1: Figures 2 and 3 are also complicated to interpret; groups 1 and 2 are not well identified.

We agree with the reviewer that the interpretation can be simplified. So, we have taken up the suggestion to display results by different groups. However, we decided to keep it separated by age due to different diet reference intakes to each age group and this will affect the adequacy of nutritional targets. Please, see changes in page 10 and 12.

Reviewer 2 Report

The article is well written and follows high scientific standards.  On the total concept I have nothing to remark.

This does not meant that I would like the article.  The article clearly says what it studies and did it, but I'm still deeply worried about the Brasilian (and World)  school eleven.  If the goal is, as the authors say, to "positively engage with the school community" I doubt this study does the opposite.  I think the very point to this is that the pupils were not at all asked how they liked the new menus.

To put it simply, what is the benefit of such spartan menus, if all capable children rush to MacDonalds or similar after the school day, as they do in many western countries.  The students should be happy with the school food.    As the authors to some extend discuss, millions of outcomes should be studied.  What is the impact of the new menus on learning outcomes, on the consumption of wood and amount of left-over food?   On the long run the menu impact on the health of the students should be studied.   

Some odd wordings also exist.  What is item "hotdog bread" in the menu.  Is it a traditional hotdog with a sausage (of what?), or is it just the bread as one could expect in this context.   If so, why would it be called "Hotdog bread"?  Please explain in more detail.

Maybe the most fatal error I think I find in page 19.  I think the authors try to mislead the reader.  If really so that "consumption of processed meat, even not red meat, is a risk factor for the development of cancer in humans, having been classified as group 1, which gathers sufficient scientific evidence" please provide references to this evidence.   And if you say in the next sentence " The consumption of unprocessed meat was classified as a
probable carcinogen in humans, being part of group 2A" you mislead people.  Who eats unprocessed meat?????????  This all gives the reader the bad and sad impression that you are not objective in your messages.

When you also refer to different national institutions, you should give the country.  For example on page 3 you speak of Institute of Medicine.  As the article studies Brazil one could expect the institute is in Brazil.  Or maybe then in UK, as many authors come from there.  But no, reference seems to be to US.  Please be more careful, there are for example hundreds of institutes of medicine in the world.

Author Response

We are grateful for the reviewers’ comments and have been through them carefully. We believe the manuscript is significantly improved thanks to recommending their changes. We list below the ways we have modified the manuscript/figures in many places, and we explain for a couple of items why we feel they are beyond the scope of our paper to address fully at this time. We refer to the reviewer as R1 in the below.

R2: The article is well written and follows high scientific standards. On the total concept I have nothing to remark. This does not meant that I would like the article. The article clearly says what it studies and did it, but I'm still deeply worried about the Brasilian (and World)  school eleven. If the goal is, as the authors say, to "positively engage with the school community" I doubt this study does the opposite. I think the very point to this is that the pupils were not at all asked how they liked the new menus.

To put it simply, what is the benefit of such spartan menus, if all capable children rush to MacDonalds or similar after the school day, as they do in many western countries. The students should be happy with the school food. As the authors to some extend discuss, millions of outcomes should be studied.  What is the impact of the new menus on learning outcomes, on the consumption of wood and amount of left-over food? On the long run the menu impact on the health of the students should be studied.  

We acknowledge the reviewer’s comment and would like to clarify that the acceptability of sustainable menus was evaluated by the team of nutritionists responsible for the National School Feeding Program. Before the food menus were fully implemented, the recipes which presented low acceptability were replaced until the whole menu was accepted by at least 85% of the students, as recommended by the legislation. These data were not included in the manuscript because the evaluation was not carried out by the research group, so there was no control over the methods and scientific rigour. Hence, the reviewer’s suggestion is interesting and indeed important, the analysis of acceptability is outside the scope of this study. A cohort is underway to monitor the health conditions of these students, which includes clinical, anthropometric, biochemical and dietary assessments. This future research will build on this work to include the analysis of acceptability, food waste, and health impact on the long run. 

R2: Some odd wordings also exist.  What is item "hotdog bread" in the menu.  Is it a traditional hotdog with a sausage (of what?), or is it just the bread as one could expect in this context.   If so, why would it be called "Hotdog bread"?  Please explain in more detail.

We acknowledge the reviewer’s comment and have made the following change overall to improve writing style: “finger rolls”. The change can be seen in pag 06 at Table1.

R2: Maybe the most fatal error I think I find in page 19.  I think the authors try to mislead the reader.  If really so that "consumption of processed meat, even not red meat, is a risk factor for the development of cancer in humans, having been classified as group 1, which gathers sufficient scientific evidence" please provide references to this evidence.   And if you say in the next sentence " The consumption of unprocessed meat was classified as a probable carcinogen in humans, being part of group 2A" you mislead people.  Who eats unprocessed meat?????????  This all gives the reader the bad and sad impression that you are not objective in your messages.

We agree with the reviewer that this could generate confusion, hence we have rephrased the text: “Among animal products, processed meat has been identified as a major risk factor for the development of cancer in humans, so that, limited evidence suggests that unprocessed red meat has a certain level of risk, and it is considered as a probable carcinogen in humans [31,32].” Changes are on page 19. 

When you also refer to different national institutions, you should give the country.  For example on page 3 you speak of Institute of Medicine.  As the article studies Brazil one could expect the institute is in Brazil.  Or maybe then in UK, as many authors come from there.  But no, reference seems to be to US.  Please be more careful, there are for example hundreds of institutes of medicine in the world.

Round 2

Reviewer 2 Report

"processed meat has been identified as a major risk factor for the development of cancer in humans, so that, limited evidence suggests that unprocessed red meat has a certain level of risk, and it is considered as a probable carcinogen in humans [31,32]" is not in line with the references that are given. There is also no scientific evidence for this key assertion. In other small details the authors have neither taken my recommendations into account.